# Home health monitoring around the time of surgery: qualitative study of patients' experiences before and after joint replacement

Sabrina Grant ,[1] Ashley W Blom,[1,2] Ian Craddock,[3] Micheal Whitehouse,[1,2] Rachael Gooberman-Hill[1,2]

[1]Musculoskeletal Research Unit, Translational Health Sciences, Bristol Medical School, University of Bristol, Bristol, UK
[2]Bristol Biomedical Research Centre, University Hospitals, Bristol NHS Foundation Trust; National Institute for Health Research, Bristol, UK
[3]Department of Electrical and Electronic Engineering, University of Bristol, Bristol, UK

**Correspondence to**
Dr Sabrina Grant;
sabrina.grant@worc.ac.uk

## ABSTRACT

**Objectives** Hip and knee replacements are common major elective surgical interventions with over 200 000 performed annually in the UK. Not all patients achieve optimal outcomes or experience problems or delays in recovery. The number of patients needing these operations is set to increase, and routine clinical monitoring is time-consuming and resource-consuming for patients and healthcare providers; therefore, innovative evaluation of surgical outcomes is needed. The aim of this qualitative study was to capture the patient experience of living with a novel home monitoring sensing system during the period around joint replacement.

**Setting** One secondary care hospital in the South West, UK.

**Participants** 13 patients (8 female, 63–89 years) undergoing total hip or knee replacement enrolled into the study.

**Design** Qualitative study with thematic analysis. The system remained in situ for up to 12 weeks after their surgery and comprised a group of low-powered sensors monitoring the environment (temperature, light and humidity) and activity of people within the home. Patients were interviewed at two timepoints: before and after surgery. Interviews explored views about living with the technology, its acceptability, as well as attitudes towards health technology.

**Results** Three main themes emerged: installation of home-sensing technology on the journey to surgery, the home space and defining unobtrusiveness and pivotal role of social support networks.

**Conclusions** Patients who agreed to the technology found living with it acceptable. A home-sensing system that monitors the environment and activity of the people in the home could provide an innovative way of assessing patients' surgical outcomes. At a time characterised by reduced mobility, functional limitations and increased pain, patients in this study relied on informal and formal supportive networks to help maintain the system through the busy trajectory of the perioperative period.

## Strengths and limitations of this study

► In-depth one-to-one interviews provided insight into patients' real experiences and views as they lived with the technology in their own homes.
► Although the sample size was small, lacked ethnic diversity and included only people willing to have technology installed in their homes, there was good diversity in age and gender and some diversity in patients undergoing hip and knee replacements.
► Use of thematic analysis enabled robust analysis of data, including focus on the acceptability of the technology in real health-related circumstances.

replacement may be provided. Numbers of these procedures are rising and continue to do so as the population ages: in 2017 alone, 91 698 primary total hip and 102 177 knee replacements were performed in England, Wales, Northern Ireland and the Isle of Man.[1 2] Having a total hip or knee replaced involves removal of the affected joint and its replacement with prosthetic implants.

Longitudinal cohort studies have shown that outcomes for hip and knee replacements vary with around 20% of knee and 9% of hip replacement patients reporting long-term pain.[3] The length of stay following hip and knee replacements has declined over time,[4] and enhanced recovery pathways can further reduce length of stay.[5] Brander *et al* reported one in eight patients still had substantial pain 1 year after surgery despite 'well-fitting' and functioning implants.[6] Wylde *et al* found that 2 years following surgery, 11% of patients thought function was the same or worse than it was preoperatively.[7]

Information technology is already woven into many aspects of patients' lives. In a health context, technology may provide the possibility for older adults with chronic conditions and complex needs to remain at home

## INTRODUCTION

For people living with osteoarthritis or other forms of joint disease that have not responded to non-operative treatments, total hip or knee

and maintain an acceptable quality of life.[8] Technologies for use at home may include 'wearable sensors' to detect changes in vital signs,[9] functional monitoring, emergency fall detection[10 11] and cognitive and sensory assistance.[12] For patients in rehabilitation following hip or knee surgery, greater use of technology (such as video-conferencing and remote monitoring) have been suggested as solutions to improving the quality of care and optimising short-term and long-term patient outcomes.[13 14] A review of research up to 2017 focusing on the use of technology in the home, remote monitoring systems and design of better environments for older people[15] indicated that, despite an increase in studies focusing on local services and equipment that are patient-centred in design, many clinicians may be reluctant to accept change due to a lack of education in this emerging field and how it affects their patients. Others have highlighted the importance of research to understand the needs and experiences of older people and how these technologies are used. Evidence from studies about the use of technology can be used to inform their improvement to technology.[16]

To ensure technologies are developed in ways that make them fit for purpose and acceptable, there is a need to understand and characterise the views of the people who have experience of using them. The Hip and Knee Study of a Sensor Platform of Healthcare in a Residential Environment (HEmiSPHERE) aimed to assess the acceptability of home monitoring systems for patients in the NHS who were undergoing hip or knee replacement.[17] We describe qualitative research within HEmiSPHERE that explores and characterises patients' experiences of this technology.

## METHODS

The HEmiSPHERE study embedded within a broader project: the Sensor Platform of Healthcare in a Residential Environment (SPHERE) is an interdisciplinary research project that has developed sensor technology to monitor home environments.[18] The SPHERE system comprises a group of low-power sensors that can continuously measure anonymised, time-stamped information about the home (eg, temperature and humidity); this includes appliance monitors to capture the use of electricity and sensors to collect information about movement through silhouettes (body outlines). The system also includes a wearable wristband, worn by the patient, collecting accelerometery information about movement within their home. Collectively, the system can measure location, activity, speed and frequency of 'sit to stand' transitions as a surrogate marker for extent of movement. Installing the system requires up to 4 hours of technicians' time, and the system requires minimal input from individuals in their homes. Patients and household members operated the system via a tablet computer that contained an operating function to pause the monitoring system and to check the battery levels of the devices. As such, it is a 'passive' monitoring system.[15] To date, the SPHERE sensor system has been installed in a total of 52 homes in the South West of England, of which 13 were homes of people undergoing hip or knee replacement and who comprise the sample for this study.[18]

### Sample

Thirteen people undergoing a total hip or knee replacement for osteoarthritis were consecutively sampled and enrolled in the study. Periodically, the sample was reviewed to assess if there was diversity in age and gender; as there was reasonable diversity, we did not adjust sampling processes as the study progressed. Participants were aged between 63 and 89 years and comprised 5 men and 8 women, with 10 undergoing hip replacement and 3 undergoing knee replacement. Demographic information about the 13 participants is displayed in table 1. All names are pseudonyms.

### Recruitment

All participants provided written consent before taking part during the initial planning home visit. Patients placed on the waiting list for a total hip or knee replacement were identified and recruited from one orthopaedic centre in the South West of England. Potential participants were mailed an information pack (invitation letter, information booklet detailing the purposes of the study, description including images of the sensor system, installation procedures, detailed information about how their data would be used and stored, and reply slip). Potential participants who had returned the reply slip were contacted by the study researcher (SG) and were invited to discuss the study. Patients were screened for eligibility at their preoperative consultation by their treating consultant and then approached in the clinic by a researcher who explained the study, provided an information booklet and invited them to take part. To widen the reach of patients undergoing joint replacement in the NHS, parallel to recruitment, the study was also discussed with an interviewer on a local radio health show broadcast.

### Eligibility criteria

Eligibility criteria included adults who could read and understand English. As the study focused on the views of people undergoing hip or knee replacement for osteoarthritis, we designed the study consent procedures for an adult population. Participants were excluded if children (16 years and under) lived in the patient's home.

### Data collection and analysis

All data collection took place within the participants' own homes. The SPHERE system was installed approximately 2 weeks before the patient's surgery date (figure 1) and continued to monitor the household before, during and up to 12 weeks after surgery.

Interviews were conducted by an experienced qualitative researcher who had a background in psychology (SG), with all patients before surgery (timepoint 1 (T1)) and approximately 2 weeks after surgery (timepoint 2 (T2)). The background of the interviewer was not

**Table 1** Participant demographics

| Pseudonym | Age at first interview (years) | Gender | THR/TKR | Cohabitation status |
|---|---|---|---|---|
| Mr Hayes | 70 | Male | THR | Lives with spouse |
| Mrs Henry | 67 | Female | THR | Lives with spouse |
| Mr Price | 85 | Male | THR | Lives with spouse |
| Mrs Evans | 85 | Female | THR | Lives alone |
| Mr Connell | 67 | Male | TKR | Lives with spouse |
| Mrs Griffiths | 63 | Female | THR | Lives with son and daughter in law* |
| Mrs Wilson | 89 | Female | TKR | Lives alone |
| Mrs Murray | 71 | Female | THR | Lives alone |
| Mrs Bailey | 73 | Female | THR | Lives with partner |
| Mrs Harrison | 65 | Female | THR | Lives with spouse and son* |
| Mrs Thompson | 75 | Female | THR | Lives with spouse |
| Mr Baker | 71 | Male | TKR | Lives with spouse |

*Adult household member.
THR, total hip replacement; TKR, total knee replacement.

disclosed to participants as the collection and analysis of the data were conducted with impartiality and openness to any type of findings. In-depth interviews using probes and prompts provided understanding of lived experiences.[19] Each interview took place in the participant's home. On occasion, a household member was present. Interviews were audio-recorded, transcribed, anonymised and imported into the qualitative data management software QSR NVivo V.11.[20] Supplementary field notes were taken after the interview. All participants' names were replaced with pseudonyms, and identifiable information was removed. Interviews lasted between 45 and 60 min, and open-ended questions followed topic guides (box 1). Participants and interviewers were not known to each other before study commencement.

Each interview at T1 began with an introduction to the aims of the interview and a discussion of their journey into joint replacement (route to referral), views about the SPHERE sensor system, household constitution and health technology usage. Interviews after surgery

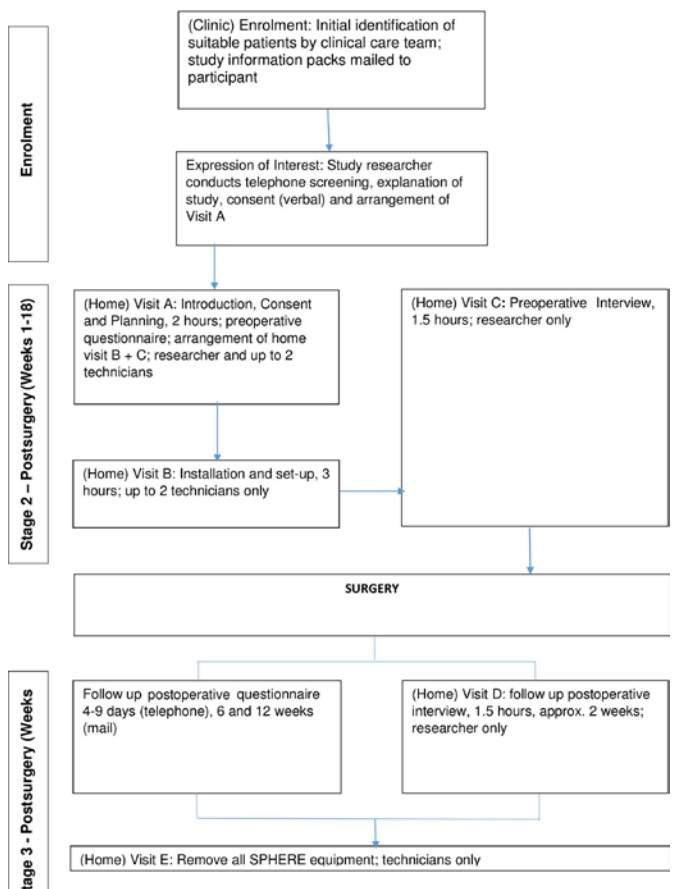

**Figure 1** Flowchart of patient through the hemisphere study. SPHERE, Sensor Platform of Healthcare in a Residential Environment.

**Box 1 Topic guides**

**Presurgery (timepoint 1)**
► Route to referral for surgery.
► People living in the household.
► Previous experience of health technology - including wearable technology, use of apps.
► Current experience and future expectations of mobility and function.
► Preparations in the household for surgery.

**Postsurgery (timepoint 2)**
► Experience of aftercare postsurgery.
► Experience of living with Sensor Platform of Healthcare in a Residential Environment (SPHERE) technology.
► Ask about the adequacy of information received about SPHERE technology.
► Explore how initial expectations of living with the SPHERE technology compared to the experience.

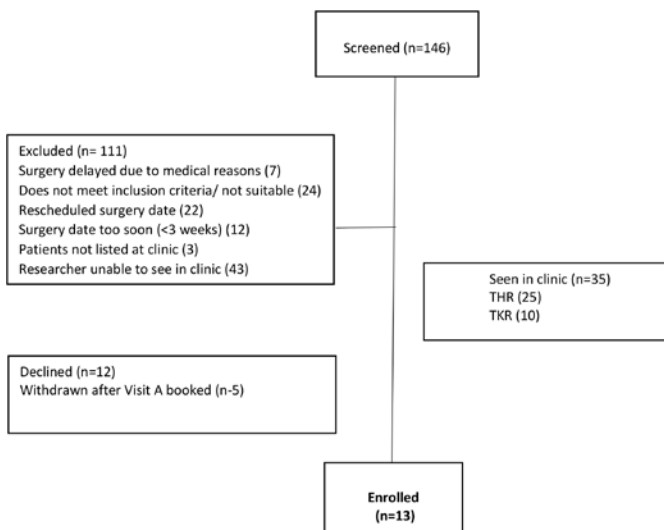

**Figure 2** Participant recruitment. THR, total hip replacement; TKR, total knee replacement.

explored care after surgery and living with the SPHERE sensor system. Using thematic analysis,[21] the researcher (SG) read and reread the data to ensure familiarity, coding inductively before sorting coded data into themes (online supplementary file 1).[19] Codes were checked for consistency and validation by a second experienced researcher in the department and the study team (SG, RG-H, MW and AWB).

We consulted a patient involvement group (described further) about the development and analyses of topic guides and used established criteria to inform our reporting of the qualitative study.[22]

### Patient and public involvement

This study was developed in collaboration with the Patient Experience Partnership in Research (PEP-R).[23] PEP-R is a patient involvement group, most of whom have had joint replacements, all of whom have had experiences of long-term pain, some after knee replacement. PEP-R provided input into research prioritisation and provided specific input into the study design, including the materials, such as recruitment documentation and interview topic guide. They also viewed and commented on feedback to the participants. SPHERE's professional advisory group was consulted on the project as a whole and the design of study materials.

### RESULTS

Of the 35 patients seen in the clinic, 12 declined to participate (concerns of living alone and managing study requirements, too burdensome and family not keen) and 5 patients and household members did not continue to full, written consent after providing their initial verbal agreement at booking visit A. The five participants did not progress to provision of full, written consent for varying reasons. Reasons included: not all members of their household provided full agreement; verbal agreement was not

followed by full formal consent; they had worries about surgery; and they had other illnesses. Of the 13 finally enrolled and completing the study, 3 were recruited by post and 1 contacted the study researcher after hearing the radio broadcast. Further details of recruitment are reported in figure 2.

We identified three main themes relating to acceptability of home monitoring technology: installation of home-sensing technology on the journey to surgery, the home space, and defining unobtrusiveness and the pivotal role of social support networks. We included illustrative quotations from the participants; all names are pseudonyms, and we indicated total hip replacement (THR) or total knee replacement (TKR) and timepoint (T1 or T2) next to each quotation.

### Theme 1: installation of 'home-sensing technology' on the journey to surgery

On route to having hip or knee replacement, a patient can expect to receive numerous letters to attend appointments with healthcare professionals (HCPs) before the day of their operation. Patients may also make plans with carers, friends or relatives about how best to recover and receive support in the weeks after surgery. This presents a critical time when anticipation of surgery may result in a period of heightened anxiety. We wanted to capture patients' experiences of being recruited and having this SPHERE sensor installed during this peak activity period.

All our patients were accepting of our approaches and the contents of the information booklet.

> Oh, it [study information booklet] was fine. I read it through, took it in, it seemed to answer anything I wanted to know (Mrs Wilson, 89 years old, TKR, T1)

> It [study visit] was okay, no it was plenty enough. Everybody's been very careful to explain every step of the way (Mr Hayes, 70 years old, THR, T1)

Providing a separate study information booklet tailored to household members gave additional assurance for others living within the household from a different perspective to the patient.

> No, I think it was fine and actually after you came the last time I said to [family members' names] 'How was that? Do you want to ask me any questions and they said, 'No, no, no. That was fine.' It was fine and then I saw them looking through the leaflet again the other night, so I think they're quite happy as well (Mrs Griffiths, 63 years old, THR, T1)

In dealing with the complex detail of the study, many felt the study information booklet and the SPHERE user guide were useful to refer to.

> Sometimes when you're talking about different things to do with this thing [the SPHERE system], you're listening but you're not really taking it in. You then think, 'What did she say about so and so?' and

then you flip through whatever (Mr Russell, 67 years old, THR, T2)

Patients and household members were shown the actual sensors (as opposed to images) and were asked to try on the wristband at the first visit. This helped them to understand the system in the context of their own home.

Well see I'm not very good at that sort of thing [technology] so what you've shown me is enough, I mean I basically know what's going on (Mrs Evans, 85 years old, THR, T2)

Patients were asked to remember some information about how to use the system. Some people found this hard to recall or to act on. For instance, remembering to charge the tablet could be hard.

I have only charged it the once I think that I can remember … I haven't bothered switching it on and off (Mr Connell, 67 years old, TKR, T1)

Participants described altruistic reasons for taking part in the study; that is, they were willing to take part in the research to provide benefit to others in the future.

I thought well it [participation in the study] would be a bit intrusive, but you know I think that everybody should do their bit to help the medical science to keep moving forward so I just thought I you know I ought to agree (Mr Hayes, 70 years old, THR, T1)

Well I hope it will help somebody because you know it's, it probably won't help me will it, I can't see how it can help me, but I can see it helping others (Mrs Evans, 85 years old, THR, T1)

Participants did not make much use of existing health technologies such as blood pressure monitors. Participants nonetheless could think of reasons how this technology could help other patients having surgery.

I thought it [participation in the study] was a good idea. I did. If it's going to help patients to recovery, it should be good. I'd hope it would help people who are having operations like this (Mr Price, 85 years, THR, T1)

Despite their preparation for major surgery, participants did not mind the level of contact required for this study before and after the operation. In the event of technical problems, participants felt they were dealt with satisfactorily but sometimes would have preferred a more rapid response.

I emailed but it was a Sunday afternoon and they [friends] were coming in the evening but I couldn't get into the thing [Genie] as I had forgot the passcode and obviously I didn't hear back until the following day when actually I rang again because nobody responded to the email on the Monday but I rang and spoke to the young lady in your office eventually. There was a little communication gap I think was all

it was really anyway, but that was fine. It was all, once I got the code I was fine (Mr Hayes, 70 years old, TKR, T1)

Participants were accepting of having to understand and comply with detailed information and study procedures despite a busy period in preparation for surgery. Thinking about the benefits for patients in the future as a result of their participation in the study appeared to be a primary motivator to allow this technology to be fitted and to monitor their activity within the home.

### Theme 2: the home space and defining unobtrusiveness

Use of the home space before and after surgery was dynamic. Before the SPHERE system was installed, participants were provided with detailed floor plans developed by the SPHERE team. These mapped areas of high traffic, where they spent the most or least time. For obvious reasons, the protocol omitted any installation of silhouette sensors in bathrooms or bedrooms.

Participants described the use of their home reflecting on the use of the space, which was often influenced by seasonal variations.

When he's [husband] I'm here [in lounge] and also, it's warmer in there. So when, you know, on a cold day it's warmer in there so we found ourselves sitting in there more … when you asked me I thought, 'I'd probably stay here most of the time'. It's only afterwards you think, 'Oh, I do go in there a lot (Mrs Thompson, 75 years old, THR, T2)

The journey after surgery sometimes gave a different perspective on the use of the downstairs space, for instance, positioning themselves in the lounge or conservatory to a preferred chair where they felt most comfortable.

One participant, Mrs Henry, described the downstairs level of her house becoming the ideal place to wash, bathe and dress. In this case, the study protocol therefore required the removal of all silhouette sensors from the downstairs level of her house. Since there were also some wireless connectivity problems in the property, it was agreed with the participant that, to save a future visit, it would be sensible to remove all the other sensors at the same time. Mrs Henry remained in the study and continued to complete the study paperwork.

I'm still getting washed and dressed at the table there because we've only got a tiny bathroom and it was not easy, it's not easy in there (Mrs Henry, 67 years old, THR, T2)

Anticipating installation of the SPHERE system in the house, some participants expressed concerns about internal damage to wall surfaces within the rooms. Although the sensor data network was wireless, some noted that the necessary power supply cabling for some of the sensors did not match their expectations, which they had expected to be entirely 'wireless'.

Well I did say to you before that I am amazed that they're not Wi-Fi or wireless. Why do they have to be cabled, I do not understand but there you are. I'm sure your technicians would know a lot more than me, it's pretty unsightly having cabling running up walls, which it is in the lounge, in the back room and in here. You know, makes what is fairly obvious even more. (Mr Hayes, 70 years, TKR, T1)

Despite concerns about aesthetics, most felt they did not have to pay much attention to the system on a day-to-day basis. Participants were given the choice to switch the system off when they went to the hospital. All decided to keep the system running during this time. Some participants wondered if they should be interacting more with the Genie, a tablet provided to the participant, which through an installed app could control the whole system, pause or delete data within certain time frames.

Sorry, just while I think about it, so all I'm doing is just looking to see when this needs charging? I'm not supposed to use the iPad for anything else? It would be nice if people could use it while they're stuck at home? It might be an encouragement for people to do the study … I don't know. I don't know what the take-up is of the study, whether it would be an extra thing that people could use it for something, I don't know. But an app could be installed on it just to do something. (Mrs Thompson, 75 years old, THR, T2)

It was a bit, I sort of wasn't quite sure what I was supposed to be doing but I'd used it just to check my batteries mainly. (Mrs Harrison, 65 years old, THR, T2)

In the immediate postoperative recovery period, some patients experienced discomfort and postoperative pain particularly in the area near the surgical wound. Adapting to the recovery phase within the household, for instance, working out how to position furniture to enable them to move, was challenging for many, particularly for patients living alone. With very little interaction with the system day-to-day and concentration on recovery, some participants described how they occasionally forgot to wear the wristband, such as after showering or when leaving the house. Some participants also described how they had not checked battery levels or had forgotten to put the wristband back on after showering. They suggested further improvements to the system, such as adding warning lights on the wristband or for warnings to be indicated clearly on the study information.

Maybe it should be in the brochure or somewhere in big letters "wear this all the time, including when you are going to bed". You just think that it's useful for recording when I walk from room to room and as going up the stairs. Because there is no recording device in the bedroom, you don't think but you obviously connect the wristband with one of those. Downstairs is where I should be wearing it. I certainly take it off before I even climbs the stairs, unless I'm going to the toilet. (Mr Russell, 67 years old, THR, T2)

Even if it could be on a sticker on the front. 'Wear this everywhere, apart from when you're in the shower' that would be good and then I would wear it, especially now, and wear it to bed. It's what people see and that's what they do. It doesn't matter about the other thing about charging it up because they'll eventually think, 'It ought to be charged by now' (Mr Connell, 67 years old, TKR, T2)

Yeah yeah and it's almost that said oh you are on that but flashed up, I don't know, like a green light or something or your battery's going or something's going on it that shouldn't it might be a good idea yeah (Mrs Evans, 85 years old, THR, T2)

Of the range of sensors within the SPHERE system, patients' experiences of the wearable wristband varied. Most patients felt the wristband was acceptable.

Do you know what, I thought I'd find it annoying. I really did think I'd find it annoying and that I would have to, because I don't even like to wear a watch, I do have a medical alert and I keep that on but I don't even like wearing a wristwatch particularly. I like to have my wrists free. Maybe because I'm not doing a lot of housework and washing, I'm not getting in the garden digging, maybe I'm not so aware of it because I'm not washing my hands a lot. So I sort of forgot about it, which surprised me a bit. So that's quite nice. (Mrs Griffiths, 63 years old, THR, T2)

However, some participants found the wristband inconvenient and sometimes took it off because of this.

Yeah, it's the left hand as well and I'm not very good with my hands. It would be better if it was a clip on or something like that. You could put it in there somewhere rather than strapping it on your wrist because it is uncomfortable as well. It's not uncomfortable as being annoying, sore or anything like that. It's just an inconvenience … I've put it in my pocket a couple of times (Mr Russell, 67 years old, THR, T2)

Two participants noted that the wristband casing was clunky and that it rattled. When trying to manage crutches postsurgery, one mentioned that the wearable caught on the crutches.

Participants felt placing of silhouette sensors in communal areas (ie, not in bathrooms or bedrooms) and knowing sensors were not recording video and sound were key system features increasing acceptability. Anonymous data collection and storage within their home, coupled with autonomy to switch the system off, increased acceptability of living with the system.

Only because [son's name] thought they'd be able to recognise us but once I explained that you couldn't recognise us, he was happy about it (Mrs Wilson, 89 years old, TKR, T1)

No, not really. I suppose the sort of cushion is that it's switch off-able if you want to, in an absolute, not that I probably won't ever but you know that it's something that you've got some control over. So even if there's lots of it about you've got control over it, isn't it, to a certain extent (Mr Baker, 71 years old, TKR, T1)

Some participants described their concern that the system was capturing information about incorrect performance of exercises or other aspects of postoperative recovery. For instance, they were concerned about their dignity as they moved around their homes unclothed on their return home from the hospital.

This is silly really but the toilet steps are hard to come from the bedroom and obviously because I get hot at night I just sleep without any clothes on and the first night I came out it was looking at me and I'd got no clothes on … that's understandable though isn't it? Of course it [the home-sensing system] doesn't see does it, but now I don't worry about it but the first night I was really panicking, I rushed back and got a dressing gown. (Mrs Evans, 85 years old, THR, T1)

No. The only time I noticed it was when I came home and thought, 'Hang on a minute' [laughs]. All my private things like washing, going to the loo, these cameras are watching me. (Mrs Henry, 67 years old, THR, T2)

### Theme 3: pivotal role of social support networks

Leading up to hip or knee replacement, patients attend numerous preoperative hospital appointments. Coupled with an anxious wait of an unfamiliar operation and unknown postoperative outcomes, patients often draw on additional support from their informal networks. While participants were fully informed that the system did not provide a realtime monitoring function, household members nonetheless felt encouraged at the prospect of taking part in a study that would 'monitor' their health and outcomes outside of a hospital environment during this unfamiliar time. Considering their increasing pain and functional decline before hip or knee surgery, many patients make use of their informal social support networks. Such support networks encouraged patients to take part in the research and to live with the SPHERE system once installed.

Well my daughter was visiting, and she said it [taking part in the study] was a great idea. (Mrs Evans, 85 years old, THR, T1)

Yes, I was. I was pleasantly surprised and when I spoke to my daughter in London, she said, 'Oh, that's great. You're going to go ahead with this, aren't you, mum?' I said yes.… Well this was charged because [daughters name] did it for me so since I've been out of hospital, I didn't know how to do it, I couldn't remember and [daughter] sort of actually worked it out. (Mrs Wilson, 89 years old, TKR, T1)

Friends, carers and family members ensured the system was working as the patient went into hospital and that wristbands were charged ready for return. Household members, or grandchildren for those living alone, took on the responsibility of checking the Genie occasionally across the 12–14 weeks. Family members were also central to addressing any technical issues; this was particularly the case during the immediate postoperative recovery period.

And [husbands' name] charged it [the wristband] up a couple of times … because the battery was going low on it (Mrs Harrison, 65 years old, THR, T2)

Participants described how grandchildren, younger friends and family members were curious about the system. Participants valued the ability to turn the system off, though none of the participants did so over the course of study participation. Knowing they could turn the system off at any point gave them some control of the system and reassurance that privacy could be maintained. One participant who rented out a room within her home to the public as a business venture spoke about how she would manage the system.

I had some [queries] last night, actually, is just, 'You may show up as you're coming through the front door.' I can't remember whether there was going to be something in the hallway or not but if people want to say yes or, 'I don't want to come,' that's fine. I'm not seeing it as stopping my business and, actually, for the first few weeks I'll shut the calendar off anyway but after that I can get them back (Mrs Griffiths, 63 years old, THR, T1)

Receiving an additional layer of support and contact from a study research team before and after the operation may have served as reassurance. Providing detailed study information about the various study visits, the technology and the data collected, including how the data would be used and stored, was also pivotal to patients agreeing to take part.

The presence of a system within the home provided intrigue and curiosity for household members, visitors, friends and family. Operating the system became a shared responsibility due to the busy period characterised by reduced mobility, functional limitations and increased pain. Patients depended on informal supportive networks to help maintain the system through the busy trajectory of surgery.

## DISCUSSION
### Main findings

This study captures patients' experiences of living and interacting with a home monitoring system in the days before and after a major operation. Overall, patients were positive about the installation and presence of the sensors before surgery. The system did not appear to interfere with their preparation for, or recovery from, hip or knee replacement.

However, with an unobtrusive sensing system comes an increased risk of patients disengaging with the system. In our study, key features of the system maintenance, such as charging the wristbands, checking the battery levels of each individual sensors and the main user-interface system (Genie), which itself required charging, were subsequently lost at times during participation.

Aesthetics and the location of the SPHERE system's sensors were also key to acceptability of this technology. Initially, some expressed concern over damage to surfaces of the walls. After surgery, most patients lived usual day-to-day activities with very little or no consideration to the sensors once installed. As the patient progressed to the recovery period and returned home from surgery, key elements of the system, such as anonymity of the data and being able to turn the system on and off, became more salient to the patient if their recovery locations within the house had changed. These features were essential in enabling the patient to retain some control over a system potentially monitoring them and the household in an unfamiliar recovery period.

In parallel to the vital support provided to the patient before surgery and in recovery by partners, carers and friends and family members, these informal support networks also assisted the patient with the SPHERE system once installed and throughout the recovery period.

The period of study participation was at a time marked by frequent visits from family, friends and carers. Any perceived intrusion or uncertainty about the presence of the system felt was mitigated by the provision of detailed information relating to the sensors, installation procedures and details about how their data would be used and stored. Such details, combined with an emphasis of anonymisation of data, were key aspects underpinning acceptability and important to understand when perceiving value within a home monitoring system. Such features translate to any long-term acceptability of 'lifestyle monitoring' technology, that is, third generation systems that have more complex capabilities and include the measurement, collection and analysis of data in the user's home.[24 25]

### Findings in relation to existing literature

Our findings build on and reflect an increasing body of research about wearable sensors and systems for home monitoring, in the context of rehabilitation. A key finding in such work has been the importance of end users' full engagement with any home-based system.[26–32] One focus group study conducted by Papi *et al*[33] of patients with osteoarthritis discussing the use of protype wearable technology indicated patients felt that they would not be able to wear the device at night. None of the patients in our study expressed any discomfort of the wearable during the night. This indicates that what patients may feel in a hypothetical situation looking at prototype technology can differ from experiencing this technology under real circumstances and serves as a strength for undertaking real-world installations of this technology with the intended user groups.

In line with other studies on technology readiness,[34 35] using any new technology may be enhanced when patients understand why the data are useful and how it may have positive impact on aspects of health. An example might be the use of technology to improve sleep, which is a common problem after a hip or knee replacement.[36] Furthermore, people must also see the value of the technology or device to use it.[32] Patients and family members could see the value and future benefits of using a home monitoring system contributing to their willingness to be involved in the study.

Participants generally felt that their own privacy was appropriately addressed by the study design but expressed more concerns relating to the privacy of friends and family entering the home, though these were mitigated by the option for participants to switch the system off at any time. Where, as here, sensor systems are designed to be respectful of the higher expectation of privacy in rooms such as bedrooms and bathrooms, studies need to consider whether patients might regularly need to wash or dress in other rooms as a consequence of their health condition. This contributes to the wider understanding of the challenges of 'in the wild' studies, which need to be sensitive to private spaces, changing personal circumstances and complicated models of consent.[37 38]

Drawing on informal networks and social support has been previously explored with patients undergoing knee replacement.[39] Although participants valued their independence, most accepted the need to rely on family friends to help them in their journey of having surgery, and this included helping them operate the home monitoring system.

Postsurgery patients felt the system could benefit from more interaction. Providing visual feedback (eg, on movement) would have rendered this an interventional study, which would have gone beyond the scope of the present study; however, adding that capability for other studies remains important future development work.

### Strengths and weakness of the study

Our use of in-depth qualitative methods provided detailed insight into the experiences and views of patients who were using home health technology around the time of surgery. One-to-one interviews enabled us to explore and probe their views in some depth, and our use of thematic analysis allowed us to deepen our understanding of the factors that underpinned its acceptability. Although our sample was relatively small, there was good age and gender diversity and some diversity in hip and knee replacements. Also, although the study only included people willing to have technology installed in their homes, we have confidence that we achieved sampling adequacy.[40] We found that no new themes emerged after the 10th participant, indicating that we had achieved data saturation and were able to interview a diverse cross section of participants in terms of age, gender and

surgery type. A potential weakness of the study is under-representation of participants across a wide range of socioeconomic groups. We noted, for instance, that the sample included only individuals who self-identified as white. The sample is reflective of national figures of the proportion of joint replacements among different ethnic groups in England.[41] Further research could include more diversity to ensure that the work is more generalisable to a range of populations and those who may have had a longer length of stay in hospital or complications. It is also likely that views about the SPHERE technology could have been different among a larger group of participants undergoing TKR. The recovery profiles of patients undergoing THR and TKR differ, and therefore these patients may have differing capacities to benefit from technology that helps to monitor recovery and provide early warning of problems that might benefit from assessment or intervention.[42]

## CONCLUSIONS

Patients who agreed to the technology found living with it acceptable, but we cannot conclude from this sample whether all people having THR and TKR would be willing now or in the future to have the system in their homes. Ultimately, that willingness could be shaped by the perceived benefit of the intervention or service that the technology was enabling, which is beyond the scope of this preliminary study.

Participants were able to conduct their preparation for surgery and recovery afterwards, but sometimes forgot to wear the wristband and to keep sensor batteries charged. Informal support networks were key to enabling them to operate the smart home technology before and after surgery. We recommend that development of smart home technology for use at the time of major healthcare intervention consider the roles of other members of a patient's social networks in the successful use of the system.

Finally, for HCPs and patients working together in consultation, continuous home monitoring may provide a useful picture of activity and function during preparation for and recovery from hip or knee replacement. The SPHERE system is currently in a development stage. Further research will identify how the system will support the delivery of future services, such as assisted living or rehabilitation.

**Acknowledgements** We thank Dr Abir Ghorayeb for double coding a portion of the transcripts and all the patients who participated in the study.

**Contributors** RG-H, IC and SG conceived the idea of a qualitative exploration. All authors contributed to the study design and participated in the analysis and writing. SG led on the writing and gathered the data. All authors commented on and approved the final version of the manuscript.

**Funding** This work was funded by an Engineering and Physical Sciences Research Council grant (grant number EP/K031910/1).

**Competing interests** None declared.

**Patient consent for publication** Not required.

**Ethics approval** Ethical approval was obtained from South West–Central Bristol Research Ethics Committee (17/SW/0121) on 22 June 2017.

**Provenance and peer review** Not commissioned; externally peer reviewed.

**Data availability statement** Anonymised data may be shared via the University of Bristol Research Data Repository. Access to the data will be restricted to ensure that data is only made available to bona fide researchers for ethically approved research projects, on the understanding that confidentiality will be maintained and after a Data Access Agreement has been signed by an institutional signatory.

**ORCID iD**
Sabrina Grant http://orcid.org/0000-0003-0148-9103

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
