## [Reviewer comments · BMJ Open]

ARTICLE DETAILS

TITLE (PROVISIONAL)	Home health monitoring around the time of surgery: a qualitative study of patients' experiences before and after joint replacement
AUTHORS	Grant, Sabrina; Blom, AW; Craddock, Ian; Whitehouse, Micheal; Gooberman-Hill, Rachael

VERSION 1 – REVIEW

REVIEWER	Professor Rebecca Jester University of Wolverhampton, UK
REVIEW RETURNED	30-Jun-2019

GENERAL COMMENTS	This is an interesting paper. It would be useful in the background information to differentiate between TKR and THR patients regarding satisfaction as THR patients are known to be more satisfied generally than TKR patients. Specifically circa 17% of TKR patients report residual pain and unmet expectations regarding function. Also it would be useful to insert a sentence or 2 regarding average LoS after TKR/THR especially with fast track/ enhanced recovery pathways leading to very short periods of hospitalisation e.g 2-3 days. Did you explore if LoS and postoperative complications influenced participants' views of the technology?
---

REVIEWER	Samantha Bunzli The University of Melbourne
REVIEW RETURNED	02-Jul-2019

GENERAL COMMENTS	Abstract: Line 8: THE aim of this study Line 9: ...THE patient experience Line 15: enrolled INTO the study Line 20: what do you mean by monitoring the environment? What aspects of the environment were monitored? Line 30-32: "It should be considered that patients depended" - this sentence doesn't sound quite right. Are you suggesting that the audience should consider that patients (more generally) depend on support networks; or that the patients in the study depended on support networks? Page 3, line 18-31 - this is a long paragraph with multiple concepts and no references Page 3. I think that there needs to be more of a link made between the argument that outcomes from TJR need to be optimised and how the authors believe that home technologies will lead to improved patient experience/ improved health outcomes post-TJR. Is the aim to better capture mobility/functional outcomes post-TKR?
--

	Is the aim to design interventions targeting mobility/function post-TKR? Page 4, line 21 -is consecutive sampling correct? Did 13 consecutive patients attending the orthopaedic clinic agree to have the technology installed in their home? (I see page 5 line 58 that this is not the case). Purposive sampling to capture diversity would be more consistent with the qualitative approach. You also want to make sure that you capture the experiences of people that can provide you with rich insights into their experiences Page 4, line 23 - I note that most participants were undergoing THR and only three undergoing TKR. Can the authors comment on the implications of this, given evidence that pain/function/satisfaction outcomes can differ between THR and TKR? Page 4, line 48 - I note the exclusion of people with children in the home how do the sensors account for other adults who may be living in the home? I understand that there is not scope for the authors to go into detail about the technology in this paper, but I think we need to know something about it. Later in the findings sections the authors refer to Genie, tablets, wristbands ... it is important that you tell the readers something about what the participants are being asked to do in this study. Page 5, line 4 - provided understanding of the lived experience. I think more accurate to say "provided insight into the lived experience" What was the role of the interviewer /data analyst in the wider study? What is the lens through which they collected/analysed this data? Interview guide - how is 'route to referral' a relevant topic in this study? Page 5, line 38 - can the authors provide us with the coding tree so we can see how codes were reduced into themes? Page 5, line 42 - I wondered what PEP-R was then see it appears in the paragraph below. Need to spell this out in the first instance. What do the authors mean when they say that the PEP-R group consulted in the development of transcripts? How did their input shape the outcomes? Page 5, line 59 - why did the five people withdraw after the visit? Page 6, line 4 - one person was recruited via radio -but radio was not described as a recruitment strategy in the methods Page 8, line 7 - privacy is a major concern with this approach. It is important to probe this further - among other concerns, is it fair that this person had to have the whole system removed because they were unable to use their bathroom as foreseen? Did other participants not raise privacy concerns? I see now further down in page 10, line 3 privacy is a concern with other participants. Reading these quotes makes it evident to me that privacy really is a major concern that has been under-addressed in this analysis.
--	---

	Page 10, line 22 - this theme also raises concerns for me. Did the participants feel coerced by their families to have the technology installed to satisfy the families desire to monitor their loved ones? Page 11 - main findings - the authors focus on the participants' concerns about damaging walls but make very little reference here to the participants' concerns about privacy (beyond a brief remark about anonymity). As a qualitative researcher, I am concerned about the role of author bias in this study. Page 12, line 17 - this reference should not come at the end of the statement about your study. Page 12, line 22 - I think the readers would also benefit from understanding how this data can improve aspects of health. In what way can it improve sleep - can you tell us more about that study? Page 12 line 31, as stated earlier, I think privacy was a major concern in this study. Withdrawing one patient is a significant issue in a study like this. The two other quotes provided in the findings are also very important. Page 12 - line 42 - more engagement - what does this mean? Page 12 - what about the fact that 12 people declined to participate and 5 withdrew after consent? Other limitations need to be addressed here as raised in my comments Page 13 line 6. I do not think you can state from this study that people undergoing TJR are willing to accept installation of a sensing system. Page 13, line 9 - Why are the authors not discussing the implications raised by the person who bathed in their living room? The technology could not cope with this apparently. Why focus on people forgetting to keep their batteries charged, but not this? Page 13, line 13 - how can this technology provide a useful picture of activity for health professionals?
--	--

REVIEWER	Shayan Bahadori Bournemouth University Orthopaedic Research Institute.
REVIEW RETURNED	05-Jul-2019

GENERAL COMMENTS	This is a well written, concise paper looking at and answering a simple question. My concern is over definition of the quality and the appropriateness of the system to the real world. Page 4 Line 12: missing 'includes' Page 4 Line 13: define quality Page 4 Line 50: Data collection and analysis - explain time of installation and also cost. Methods: What sensors were used, how many were used. perhaps an image or illustration of the system. how did you ensure sensors were calibrated and did any of the participants use any similar system/activity monitor before. Page 8 line 7: was this person included in trial? Discussion: How was the system intended to provide outcome data. Address, topics on accuracy, reliability and repeatability. Is the cost effective?
--

VERSION 1 – AUTHOR RESPONSE

Reviewers' Comments to Author:

Reviewer: 1

3) This is an interesting paper. It would be useful in the background information to differentiate between TKR and THR patients regarding satisfaction as THR patients are known to be more satisfied generally than TKR patients. Specifically circa 17% of TKR patients report residual pain and unmet expectations regarding function. Also it would be useful to insert a sentence or 2 regarding average LoS after TKR/THR especially with fast track/ enhanced recovery pathways leading to very short periods of hospitalisation e.g 2-3 days. Did you explore if LoS and postoperative complications influenced participants' views of the technology?

Thank you for these helpful suggestions about adding more detail about length of stay (LoS), satisfaction and outcomes after TKR and THR. We have added some detail into the introduction to provide further context. We have added that outcomes for hip and knee replacement vary, with around 20% of knee and 9% of hip replacement patients reporting long-term pain. We have also added comment about length of stay in hospital. As length of stay varies according to whether a patient is treated under an ERAS pathway or has complications or co-morbidities, we have added some average figures for context. However, we did not investigate the relationship between length of stay, complications and views of technology. This was because the number of patients who experience complications of long length of stay is relatively small, and as a prospective study doing so would have necessitated a larger sample size and probably a cohort study with a nested qualitative study. Future work could consider this, and we have added a suggestion about this to the discussion (page 13, paragraph 5).

Reviewer: 2

4) The reviewer notes some typographical errors. We have corrected these, which are shown as track changes throughout the manuscript. Our apologies for these.

5) The reviewer asks that we clarify what is meant by monitoring the environment in the abstract. We have added this detail (abstract, line 16).

6) The reviewer asks that we revise the sentence starting: "It should be considered that patients depended" they write that this sentence doesn't sound quite right. Are you suggesting that the audience should consider that patients (more generally) depend on support networks; or that the patients in the study depended on support networks?

Thank you, we have revised this sentence for clarity, as the word 'considered' is not quite right here.

7) The reviewer suggests that on Page 3, line 18-31 the paragraph is long and is not referenced.

Thank you. We have combined suggestion 7) and 8) and taken out this paragraph. To emphasise the link between technology and improvements in patient health outcomes we include a sentence and references to substantiate the need for technology to improve rehabilitation after surgery and the monitoring of post-operative patient outcomes. This can be found on page 3 paragraph 3.

8) Page 3. I think that there needs to be more of a link made between the argument that outcomes from TJR need to be optimised and how the authors believe that home technologies will lead to improved patient experience/ improved health outcomes post-TJR. Is the aim to better capture mobility/functional outcomes post-TKR? Is the aim to design interventions targeting mobility/function post-TKR?

Thank you for your recommendation. Please see our response to suggestion 7.

9) Page 4, line 21 -is consecutive sampling correct? Did 13 consecutive patients attending the orthopaedic clinic agree to have the technology installed in their home? (I see page 5 line 58 that this is not the case). Purposive sampling to capture diversity would be more consistent with the qualitative approach. You also want to make sure that you capture the experiences of people that can provide you with rich insights into their experiences.

Thank you. Patients were recruited through consecutive sampling approaches and as sampling continued the diversity was checked to ensure that there was reasonable diversity in terms of age and gender. We appreciate that the sample for this type of work will always entail an element of selectivity as only those individuals willing to have technology installed in their homes took part, and this is a weakness of the study. We have therefore clarified the sampling and added this as a weakness that future research could address. Page 13 paragraph 3.

10) Page 4, line 23 - I note that most participants were undergoing THR and only three undergoing TKR. Can the authors comment on the implications of this, given evidence that pain/function/satisfaction outcomes can differ between THR and TKR?

Thank you, yes most participants were undergoing TKR and only three undergoing THR. As the sample was small we did not seek to compare the groups, but given the different outcomes after TKR and THR we have added reflection within the strengths and weaknesses section of the discussion about possible differences in the discussion. These include that patients having either TKR or THR may have capacity to benefit from technology that helps to monitor recovery and provide early warning of problems that might benefit from assessment or intervention (page 13, first paragraph).

11) Page 4, line 48 - I note the exclusion of people with children in the home how do the sensors account for other adults who may be living in the home?

Some of the sensors (such as silhouette sensors, wearables) produce data that can itself be attributed to a specific participant, others do not (such as room humidity sensors) – it is for this reason that all adults living in the property are consented (not just the patient). We're sorry that the original paper is not clear in that regard but the results section, page 6 has now been amended to make this clear.

12) I understand that there is not scope for the authors to go into detail about the technology in this paper, but I think we need to know something about it. Later in the findings sections the authors refer to Genie, tablets, wristbands ... it is important that you tell the readers something about what the participants are being asked to do in this study.

Thank you, in addition to the references provided to papers describing the technology, we have added some edits to the first paragraph of the methods section (Page 4, first paragraph) about the sensors, including information about the data the sensors were collecting. This also acts as a response to comment 35 by Reviewer 3. (Page 4, first paragraph) about the sensors and the information they provide.

13) Page 5, line 4 - providedED understanding of the lived experience. I think more accurate to say "provided insight into the lived experience"

Thank you, we have made this change.

14) What was the role of the interviewer /data analyst in the wider study? What is the lens through which they collected/analysed this data?

The interviewer was an experienced qualitative researcher with a background in psychology. The collection and analysis of the data was conducted with impartiality and openness to any type of findings. We have added this information to the methods (page 5 paragraph 2)

15) Interview guide - how is 'route to referral' a relevant topic in this study?

Thank you. Participants were asked about their routes to referral to provide context about their journey into joint replacement. Edits made (page 5, last para)

16) Page 5, line 38 - can the authors provide us with the coding tree so we can see how codes were reduced into themes?

Yes, we include a coding tree in the appendix

17) Page 5, line 42 - I wondered what PEP-R was then see it appears in the paragraph below. Need to spell this out in the first instance. What do the authors mean when they say that the PEP-R group consulted in the development of transcripts? How did their input shape the outcomes?

Thank you, you are correct that PEP-R is mentioned and then described later. We have edited this section for clarity to explain that the patient involvement group were consulted (page 6 first para)

18) Page 5, line 59 - why did the five people withdraw after the visit?

Thank you, we have added that the five people decided not to take part after the first visit at which they provided their verbal consent. This was for varying reasons, including: not all members of the household would provide their full agreement; patients changed their decision to participate after initial verbal agreement, concerns about their surgery, and other illness. Page 6, paragraph 3.

19) Page 6, line 4 - one person was recruited via radio -but radio was not described as a recruitment strategy in the methods.

Thank you for this comment. We have made edits to the methods section (page 4, para 3) to clarify our methods of expanding awareness of the study through a local radio broadcast.

20) Page 8, line 7 - privacy is a major concern with this approach. It is important to probe this further - among other concerns, is it fair that this person had to have the whole system removed because they were unable to use their bathroom as foreseen? Did other participants not raise privacy concerns? I see now further down in page 10, line 3 privacy is a concern with other participants. Reading these quotes makes it evident to me that privacy really is a major concern that has been under-addressed in this analysis.

Thank you. We have made some edits to this section to provide greater clarity that silhouette sensors were not installed in private areas such as bathrooms and bedrooms. We have edited this sentence to add detail about why the system was removed.

Privacy is certainly an important issue with the study and was a major focus of PPI, of the study design, ethical approval and all our conversations with potential participants. Addressing privacy was the reason that participants themselves were given the ability to pause or delete data collection at any time. As described in the text by participants Mrs Wilson and Mr Baker, these steps seem to have prevented privacy becoming a particular problem for the study. Even in the unusual case of Mrs Henry, it was not the dominant reason for the system being removed. We have also added text to the discussion to address privacy concerns when designing a study of this nature. (Page 12, paragraph 8).

21) Page 10, line 22 - this theme also raises concerns for me. Did the participants feel coerced by their families to have the technology installed to satisfy the families desire to monitor their loved ones?

Thank you, we do not think that patients were coerced since they were fully-aware that the technology did not implement any function that would allow families to monitor loved ones. Patients were fully informed about the system's capabilities and limitations and therefore made their own decisions about the study. We have made sure that this is clearer by editing some of the text in this theme. (Page 11, first paragraph).

22) Page 11 - main findings - the authors focus on the participants' concerns about damaging walls but make very little reference here to the participants' concerns about privacy (beyond a brief remark

about anonymity). As a qualitative researcher, I am concerned about the role of author bias in this study.

Thank you, please see the response comment to point 14 above. The interviewer was an experienced qualitative researcher with a background in psychology. The collection and analysis of the data was conducted with impartiality and openness to any type of findings. We have added some additional text to this effect into the methods (page 5 paragraph 2).

23) Page 12, line 17 - this reference should not come at the end of the statement about your study.

Thank you, we have removed the reference.

24) Page 12, line 22 - I think the readers would also benefit from understanding how this data can improve aspects of health. In what way can it improve sleep - can you tell us more about that study?

Thank you. We have made some edits within the sentence explaining what might a sleep study might look like and that this forms part of some of the authors' further research.

25) Page 12 line 31, as stated earlier, I think privacy was a major concern in this study. Withdrawing one patient is a significant issue in a study like this. The two other quotes provided in the findings are also very important.

Thank you, though in fact no patients withdrew from the study (they continued with questionnaire data to the end of the study). As stated in the response to point 20, privacy was carefully addressed through PPI and specific tools were made available to mitigate privacy concerns - feedback from participants in the paper indicates that they understood those tools and found them helpful. The withdrawal of sensors from one property was in fact done as a result of following our own study protocol and not as a result of a participant raising a privacy concern. We apologise if this was not previously clear and hope that the new text is helpful.

Attitudes to privacy (and indeed trust) do vary from person to person and we agree that there is a great deal more work to be done on this aspect, however in this study privacy was not a particularly dominant theme in the feedback from participants.

26) Page 12 - line 42 - more engagement - what does this mean?

Thank you, we have changed the word engagement to 'interaction' for clarity.

27) Page 12 - what about the fact that 12 people declined to participate and 5 withdrew after consent? Other limitations need to be addressed here as raised in my comments.

Thank you. We have added a brief amount of detail into the methods section about reasons for non-participation, and reasons why some people did not progress from verbal to full, written consent. However these are brief and relate only to stated reasons, as we did not have consent ethical approval to collect or report reasons for non-participation in this in detail (page 6, first paragraph).

28) Page 13 line 6. I do not think you can state from this study that people undergoing TJR are willing to accept installation of a sensing system.

Thank you, we agree that this is probably an overstatement of the findings. We have therefore edited the discussion and the abstract's conclusions so that it is clearer that those who agree to the technology find it acceptable to live with, but that we do not yet know whether all people having THR or TKR would be willing now or in the future to have the system in their homes. (Abstract line 32, page 2)

29) Page 13, line 9 - Why are the authors not discussing the implications raised by the person who bathed in their living room? The technology could not cope with this apparently. Why focus on people forgetting to keep their batteries charged, but not this?

Thank you, we have added discussion about this, which relates to concerns about privacy as above (page 12, paragraph 8) and in the findings (Page 8, paragraph 3).

30) Page 13, line 13 - how can this technology provide a useful picture of activity for health professionals?

Thankyou. We believe that the data collected from this technology could help patients and health professionals to work together in consultation with this continuous home monitoring data before and after surgery. This forms some further work which the authors are currently exploring. We feel this is addressed in the sentence however we have included the word 'consultation' within the sentence. (Page 14, last paragraph)

Reviewer: 3

31) This reviewer writes that this is a well written and concise paper. They also wonder about the 'definition of the quality and the appropriateness of the system to the real world.'

We thank the reviewer for their positive comment. To address the comments about appropriateness we have revised the discussion's last sentence to make it clearer that the SPHERE system is in a development stage, and that it will deliver future benefits when the technology is ready.

32) Page 4 Line 50: Data collection and analysis - explain time of installation and also cost. Later, the reviewer also asks about economic analysis.

Thank you for asking this question, the installation took four hours of technician time at no cost to participants. We have added this information (page 4, paragraph 2). We have not conducted an economic analysis within this study, but future work (for instance any RCT) would do so.

33) Methods: What sensors were used, how many were used. perhaps an image or illustration of the system. How did you ensure sensors were calibrated and did any of the participants use any similar system/activity monitor before.

Thank you for asking about the sensors, we have added some more detail about the sensors in the methods sections so that it is clearer to readers what they were and what information they were collecting. We have though not added fine detail about calibration, which we think best suited to a technical audience. None of the participants were using similar technology at home before taking part. (page 4, paragraph 2)

34) Page 8 line 7: was this person included in trial?

Though the system was removed the participant wished to remain in the wider study for the collection of questionnaire data. We have included edits in this section to clarify this point. (Page 8 para 3)

35) Discussion: How was the system intended to provide outcome data.

The sensors combined was anticipated to provide various types of outcome data (e.g. temperature, light, humidity, activity, location) providing a unique way of measuring activity within the home as it applies to the recovery of patients undergoing joint replacement. We feel that this is addressed throughout the paper.

36) Address, topics on accuracy, reliability and repeatability.

To ensure our qualitative methods were robust and analysis accurate, we took several steps within our work such as double coding of the transcripts. This is described in the methods section (Page 5, paragraph 3).

This reviewer has noted some typographical corrections, which we have made

Page 4 Line 12: missing 'includes' - corrected

Page 4 Line 13: define quality – corrected, by quality we mean speed and frequency of 'sit to stand' transitions as a surrogate marker for extent of movement.

VERSION 2 – REVIEW

REVIEWER	Rebecca Jester Professor of Nursing Institute of Health Faculty of Education, Health and Wellbeing University of Wolverhampton United Kingdom
REVIEW RETURNED	03-Oct-2019
GENERAL COMMENTS	Thank you for diligently addressing the reviewers' feedback.